# Characterization of Nuclear and Magnetic Structures of Wolframite-Type MgReO$_4$ and ZnReO$_4$

U. Miniotaite [1*], O.K Forslund[2],[3] E. Nocerino [4,5] Y. Ge [1], F. Elson [1], M. Sannemo [6], H. Sakurai [7], J. Sugiyama [8], H. Luetkens [5], T. Ishigaki [9], T. Hawai [9], R. Palm [10], C. Wang [5], Y. Sassa [1], D. W.Tam [1], M. Månsson [1**]

**1** Department of Applied Physics, KTH Royal Institute of Technology, SE-106 91 Stockholm, Sweden

**2** Physik-Institut, Universität Zürich, Zürich, Switzerland

**3** Department of Physics and Astronomy, Uppsala University, Uppsala, Sweden

**4** Department of Chemistry, Stockholm University, Stockholm, Sweden

**5** PSI Center for Neutron and Muon Sciences CNM, Villigen PSI, Switzerland

**6** Department of Materials and Environmental Chemistry, Stockholm University, Stockholm, Sweden

**7** National Institute for Materials Science (NIMS), Namiki Tsukuba Ibaraki, Japan

**8** Neutron Science and Technology Center, Comprehensive Research Organization for Science and Society (CROSS), Tokai, Japan

**9** Neutron Industrial Application Promotion Center, Comprehensive Research Organization for Science and Society (CROSS), Tokai, Japan

**10** Institute of Chemistry, University of Tartu, Tartu, Estonia

* ugnem@kth.se ** condmat@kth.se

March 26, 2025

## Abstract

**We utilized high-pressure methods to synthesize the oxides AReO$_4$ (A = Zn, Mg) and characterized their crystal structures as monoclinic wolframite-type. By combining muon spin spectroscopy ($\mu^+$SR) with DFT calculations for muon stopping sites, we identify two possible magnetic spin structures for both compounds: $\Gamma_3$ with the propagation vector k $= (0, 1/2, 0)$ and $\Gamma_4$ with k $= (0, 0, 0)$. In both cases, the magnetic moments are canted from the principal axes within the $ac$-plane. The ordered moment of the proposed structures is $0.29(5)\mu_B$ for $\Gamma_3$ and $0.25(8)\mu_B$ for $\Gamma_4$. The low moment is consistent with the absence of a magnetic contribution to the neutron powder diffraction (NPD) spectra. Bond valence sum (BVS) analysis supports the oxidation state of Re being Re$^{6+}$ in the compounds and we suggest that a combination of $t_{2g}$ orbital splitting due to spin-orbit coupling (SOC) and $d$-$p$ orbital hybridization is responsible for the strongly suppressed ordered magnetic moment.**

# 1   Introduction

Transition metal oxides play a fundamental role in condensed matter physics due to their diverse and unique states, such as high-temperature superconductivity [1, 2], exotic magnetic phases [3–5], multiferroics [6,7] and metal-insulator transitions [8,9]. This family also show their importance in applications, e.g., energy storage (rechargeable batteries) [10–12] and catalysis [13,14]. Their wide variety of properties are attributed to variations in oxygen bonding and the nature of the unfilled $d$ electron shells [15].

Rhenium, the last transition metal to be discovered in 1925 [16], has attracted interest for its broad range of oxidation states (-3 to +7) [17]. At the higher end of oxidation, we find the rhenium oxides and halides. Rhenium oxides often stabilize in the perrhenate structure in which oxygen atoms form a tetrahedral coordination around the Re atoms [18–20]. Some well-known $AReO_4$ perrhenates, where A is a metallic cation and Re is in the +7 oxidation state, include $AgReO_4$ [21] and $KReO_4$ [22], both utilized in electrochemical applications. However, there are only a few reports of such compounds with Re in a lower oxidation state [23–26].

In the 1970s, Sleight *et al.* synthesized a series of $AReO_4$ oxides which were found to have rutile (A = Al, Fe, Ga) and wolframite-type (A = Mg, Mn, Zn) structures. The goal was to stabilize rhenium in lower oxidation states [23] and to combine localized $3d$ electrons with delocalized $5d$ electrons, potentially leading to novel electronic and magnetic properties. Based on bond length estimates from lattice parameters, most rutile compounds were assigned a $Re^{5+}$ oxidation state, while the wolframite-type oxides were assigned $Re^{6+}$ [23]. Notably, $Re^{6+}$ compounds exhibit intriguing magnetic properties,

including weak persistent ferromagnetism (FM) above room temperature [27–29] and pronounced magnetic anisotropy [30, 31]. The wolframite structure, consists of edge-sharing, distorted octahedral coordination of oxygen around Re atoms. The origin of octahedral distortions in rhenium oxides has been attributed to Jahn-Teller (JT) effects [32], Re-Re bonding [33], or variations in ionic radii [34].

Among these compounds, Bramnik *et al.* [35] have characterized the wolframite structure of $MnReO_4$ and, through magnetic susceptibility measurements, identified an antiferromagnetic (AFM) component with a Néel temperature, $T_N$, of 280 K, coexisting with weak FM below the Curie temperature of 225 K. Mn in $MnReO_4$ possesses unpaired electrons, leading to strong magnetic moments which obscure the contribution from $Re^{6+}$. In contrast, neither Mg in $MgReO_4$ nor Zn in $ZnReO_4$ has unpaired electrons, meaning that any magnetism in these compounds originates solely from $Re^{6+}$, making them ideal candidates for studying its intrinsic magnetism.

In this study, we investigate the magnetic properties of the wolframite-type rhenium oxides $AReO_4$ (A = Mg, Zn). To characterize their structure, we conduct neutron powder diffraction (NPD). The diffraction patterns show no visible magnetic contributions for either sample at 7 K, despite clear magnetic transitions observed in both AC magnetic susceptibility (ACMS) and muon spin spectroscopy ($\mu^+$SR), indicating a weak magnetic moment.

Our preliminary analysis of $MgReO_4$ using $\mu^+$SR identified AFM ordering below $T_N = 83$ K [36]. At base temperature ($T = 2$ K), two muon precession frequencies were observed, originating from two distinct muon stopping sites. However, above $T = 60$ K, only one precession frequency is present in the data. This was initially speculated to be a result from spin canting, causing both muon sites to experience the same magnetic moment.

Here, we provide a more detailed analysis of the $MgReO_4$ $\mu^+$SR data [36] and extend the study to $ZnReO_4$. We revisit the $MgReO_4$ $\mu^+$SR data reported in [36] with the advantage of a refined crystal structure from complementary NPD data, gaining additional insights into the system's magnetic properties. Using a combination of DFT, $\mu^+$SR and NPD, we identify two possible magnetic structures for both compounds consistent with zero field (ZF) $\mu^+$SR data. By examining the magnetic structure under canting conditions, we rule out spin canting as the cause of the reduction in observed muon precession frequencies. Bond valence sum (BVS) analysis supports the $Re^{6+}$ oxidation state, ruling out wrongful oxidation state as the cause of the low observed ordered moment. Instead, our results suggest that strong spin-orbit coupling (SOC) in the $5d^1$ system splits the $t_{2g}$ orbitals, with the lowest energy level being a non-magnetic quadruplet state, causing the suppressed magnetic moment.

## 2 Experimental Methods

High-quality powder samples of $ZnReO_4$ and $MgReO_4$ were synthesized at the National Institute for Materials Science (NIMS), Ibaraki, Japan, using a high-pressure synthesis technique. The material was subjected to a pressure of 6 GPa for one hour, at a temperature of 1300 °C. As the samples are sensitive to humidity, in particular $ZnReO_4$, all sample preparations were conducted in an Argon glove box.

To ensure sample quality, due to the high humidity sensitivity, $ZnReO_4$ was checked using ACMS (Appendix A). ACMS measurements were conducted using the Quantum Design PPMS DynaCool. The sample was compressed inside a capsule and sealed with kapton tape.

NPD was used to characterize the structures. The NPD patterns were collected at high-resolution time-of-flight (TOF) diffractometer iMATERIA at J-PARC [37], Ibaraki, Japan. Approximately $0.5\,\mathrm{g}$ of each sample was sealed in a cylindrical vanadium can (diameter 5 mm) using aluminium screws and indium wire. For structural refinement, the high-Q detector banks; back scatter (BS) with resolution $\Delta d/d = 0.15\%$ and $90°$ SE detector bank with resolution $\Delta d/d = 0.45\%$, were used. The diffraction spectra were refined together using `FULLPROF SUITE` [38]. We also attempted to characterize the structures using X-ray diffraction (XRD), but due to high X-ray absorption of rhenium and reactions with grease, this was unsuccessful (see Appendix B).

The General Purpose surface muon beamline (GPS) at the Paul Scherrer Institute (PSI) in Villigen, Switzerland, was used to conduct the $\mu^{+}$SR experiments. Approximately 200 mg each of $ZnReO_4$ and $MgReO_4$ was sealed with Kapton tape inside an aluminum mylar-tape envelope (thickness $\approx 50\,\mu\mathrm{m}$). The sample was then mounted on a low-background copper fork and inserted into a He-4 flow cryostat, allowing for a temperature range of $T = 2\,\mathrm{K}$ - $300\,\mathrm{K}$.

The $\mu^{+}$SR measurements were performed in different magnetic field configurations, defined by the field alignment relative to the initial muon spin polarization: zero field (ZF), transverse field (TF), and longitudinal field (LF). In ZF, no external field is applied. In TF, the field is perpendicular to the initial muon spin polarization, while in LF, it is applied parallel to the polarization.

ZF spectra were collected for $ZnReO_4$ between $T = 2\,\mathrm{K}$ and $140\,\mathrm{K}$, and for $MgReO_4$ between $T = 2\,\mathrm{K}$ and $100\,\mathrm{K}$. TF measurements were performed using a magnetic field of $B = 50\,\mathrm{G}$, while LF spectra were measured with $B = 10\,\mathrm{G}$ and $B = 20\,\mathrm{G}$. The data was analyzed using the `musrfit` software [39].

The electrostatic landscapes were calculated using the pseudopotential-based plane-wave method as implemented in `QUANTUM ESPRESSO` [40]. Based on the obtained muon site candidates, the local magnetic fields were evaluated for different spin configurations using the package `muesr` in Python [41].

## 3   Results

### 3.1   Neutron Powder Diffraction

NPD was used to determine the crystal structure and atomic positions in $MgReO_4$ and $ZnReO_4$. A comparison of the diffraction patterns collected above $T_{\mathrm{N}}$ and at $7\,\mathrm{K}$ shows no evidence of a structural transition or appearance of a magnetic contribution to the pattern in any of the detector banks [insets Fig. 1 (a)-(b)]. This may on a first glance contradict the proposition of the samples being magnetically ordered. However, as we shall show below (Sec. 3.3), the absence of a magnetic contribution is consistent with small ordered moments which can also explain our $\mu^{+}$SR results. Low magnetic moments have resulted in the absence of magnetic peaks in NPD in other $Re^{6+}$ systems [42, 43].

The crystal structure refinement was performed using the patterns collected at $7\,\mathrm{K}$ to maximize the accuracy of the magnetic dipole field calculations (Sec. 3.3). The measured patterns [Fig. 1 (a)-b)] were refined to a monoclinic wolframite-type structure with the space group P2/$c$ (# 13) [44]. The refined lattice parameters [Tab. 1] are close to the values found by Sleigh *et al.* [23]. The structure [Fig. 1 (c)-(d)] contains edge-sharing distorted octahedral coordination of O surrounding the Re atoms. The octahedra form a zig-zag pattern along the $c$-axis and sandwiches the Mg/Zn atoms along the $a$-axis. In the distorted octahedra of both samples, the O-Re-O bond angles range from $76°$ to $101°$,

deviating from the ideal 90°. Additionally, the octahedra are compressed along the $a$-axis and elongated along the $c$-axis. The Re center is also displaced along the $b$-axis, leading to varying Re-Re distances along this direction. These distortions are very similar in both compounds and resemble $MnReO_4$, which has bond angle variation of 81.9° to 101.9° [35].

| MgReO$_4$ | |
|---|---|
| Crystal structure | monoclinic P2/$c$ |
| $a, b, c$ (Å) | 4.67703, 5.57462, 5.01010 |
| $\beta$ (°) | 91.9017 |
| $V$ (Å$^3$) | 130.5547 |
| Mg $(x, y, z)$ | (0.5, 0.678, 0.25) |
| Re $(x, y, z)$ | (0, 0.169, 0.25) |
| O1 $(x, y, z)$ | (0.217, 0.111, 0.943) |
| O2 $(x, y, z)$ | (-0.258, 0.627, 0.608) |
| | |
| Refinement | |
| $\chi^2$ | 0.3543 |

| ZnReO$_4$ | |
|---|---|
| Crystal structure | monoclinic P2/$c$ |
| $a, b, c$ (Å) | 4.68845 , 5.60300, 5.02260 |
| $\beta$ (°) | 91.2713 |
| $V$ (Å$^3$) | 132.2557 |
| Zn $(x, y, z)$ | (0.5, 0.686, 0.25) |
| Re $(x, y, z)$ | (0, 0.166, 0.25) |
| O1 $(x, y, z)$ | (0.214, 0.115, 0.947) |
| O2 $(x, y, z)$ | (-0.250, 0.636, 0.613) |
| | |
| Refinement | |
| $\chi^2$ | 0.1231 |

Table 1: Crystal structure parameters of $ZnReO_4$ and $MgReO_4$ from refinement of neutron powder diffraction data.

Both samples exhibit impurity peaks around $Q$ = 1.8 - 2 Å, as well as dispersed peaks at higher $Q$ in the spectra. The impurities could not be refined to any known compound containing O, Re, and/or Mg/Zn, and the regions containing these peaks were therefore excluded from the refinement. Since the impurity peaks remain visible above $T_N$, they do not seem to be of magnetic origin. They might represent a mix of multiple compounds formed during the high-pressure synthesis process. This impurity could also be accounted for as a paramagnetic, exponentially decaying signal in the $\mu^+$SR data (Sec. 3.2).

## 3.2  Muon Spin Spectroscopy

To gain information about the magnetic structures, we conduct a $\mu^+$SR study on both $MgReO_4$ and $ZnReO_4$ as a function of temperature. The ZF $\mu^+$SR time spectra of $ZnReO_4$ and $MgReO_4$ is presented in Fig. 2. The ZF time spectra at low temperatures reveals multiple oscillations in both samples. The presence of oscillations indicates magnetic ordering and multiple oscillations suggests multiple muon stopping sites inside the crystal structure. In a powder, two thirds of the internal magnetic field components are expected to be perpendicular to the initial muon spin polarization (causing oscillations), whereas

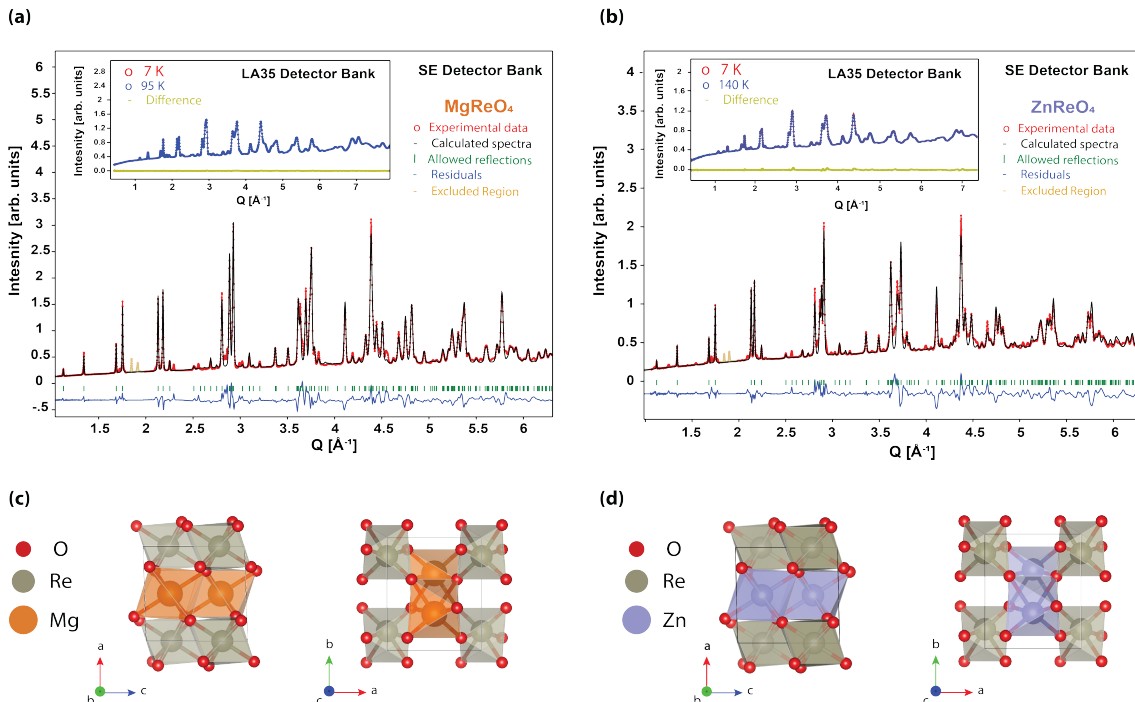

Figure 1: Measured neutron powder diffraction (NPD) patterns, refined spectra, allowed reflections, and residuals for data collected at 7 K from the SE detector bank for **(a)** MgReO$_4$ and **(b)** ZnReO$_4$. Regions excluded due to impurities are colored in orange. The insets show the differences between patterns obtained from the low-Q detector bank LA35 at 7 K and 95 K for MgReO$_4$ **(a)** and 140 K for ZnReO$_4$ **(b)**. The monoclinic wolframite structures of **(c)** ZnReO$_4$ and **(d)** MgReO$_4$ illustrate the oxygen octahedra surrounding Re atoms in the *a-b* plane and *a-c* plane.

one third is expected to be parallel to it (causing an exponentially decaying tail). We verify the asymmetry ratios between the tail and AFM oscillations in Appendix C.

As the temperature increases above $T_N$, the oscillations evolve to a static Gaussian Kubo-Toyabe (KT) multiplied by an exponential type depolarization [45], describing muon spin relaxation in a system of randomly oriented static spins within the muon lifetime. We use LF measurements to confirm KT to be the most suitable polarization function (Appendix D). Furthermore, the spins are already decoupled at a small applied magnetic field of 20 Gauss, indicating that the spin distribution above $T_N$ arises from nuclear moments, which appear static within the muon lifetime. To account for all these processes within the considered temperature range, the ZF data for both samples was fitted with the following polarization function:

$$
\begin{aligned}
A_0 P_{\mathrm{ZF}}(t) = & A_{\mathrm{AF1}} \cos\left(2\pi f_{\mathrm{AF1}} + \frac{\pi\phi}{180}\right) e^{-\lambda_{\mathrm{AF1}} t} \\
& + A_{\mathrm{AF2}} \cos\left(2\pi f_{\mathrm{AF2}} + \frac{\pi\phi}{180}\right) e^{-\lambda_{\mathrm{AF2}} t} \\
& + A_{\mathrm{KT}} G^{\mathrm{SGKT}}(t, \Delta_{\mathrm{KT}}) e^{-\lambda_{\mathrm{KT}} t} \\
& + A_{\mathrm{tail}} e^{-\lambda_{\mathrm{tail}} t} \\
& + A_{\mathrm{Im}} e^{-\lambda_{\mathrm{Im}} t},
\end{aligned}
\tag{1}
$$

where $P_{\mathrm{ZF}}(t)$ is the muon polarization in ZF, $A_i$ the corresponding asymmetry, $\lambda_i$ the

damping, $\nu_i$ and $\phi_i$ the frequency and phase of the oscillations, and $\Delta_{\mathrm{KT}}$ the Gaussian distribution width of the KT and $\lambda_{\mathrm{KT}}$ the relaxation rate of the KT. In addition to the AF1, AF2 and KT contributions, the polarization has two exponentially decaying terms; a very slowly decaying tail, $\lambda_{\mathrm{tail}}$ coming from the magnetic moments parallel to the initial muon spin polarization, and a decaying signal $\lambda_{\mathrm{Im}}$ attributed to impurities in the sample. A similar impurity phase has been found in both compounds and it has previously been speculated for $MgReO_4$ to arise from Re impurities [36]. In the fitting, the impurity fraction $A_{\mathrm{Im}}$ is fixed to a value determined from TF measurements at low temperature (see Appendix E). Furthermore, the phase of AF oscillations is theoretically $\phi = 0$. However, uncertainty in the muon implantation time, wide internal field distributions, and incommensurate (IC) order can cause $\phi \neq 0$, and therefore, it was treated as a fitted parameter. We confirm, however, that the fitted value remains close to zero at base temperature, and the phase is constrained to be the same for the two muon sites, AF1 and AF2.

### 3.2.1   MgReO$_4$

The raw data for the $MgReO_4$ $\mu^+$SR analysis is the same as in Ref. [36]. In this work, we reanalyze it with a greater focus on determining a detailed spin structure and comparing it with $ZnReO_4$. The ZF muon spectra and temperature dependence of the polarization function (Eq. (1)) fitting parameters of $MgReO_4$ are shown in Fig. 2. Above $T_{\mathrm{N}}$, $A_{\mathrm{KT}}$ dominates, as expected for a paramagnetic sample. As the temperature is lowered, this asymmetry decreases in favor of $A_{\mathrm{AF1}}$, $A_{\mathrm{AF2}}$ and $A_{\mathrm{tail}}$ indicative of a magnetic ordering with the transition temperature $T_{\mathrm{N}} = 83.35(2)$ K with the error being a statistical error in the fit (Appendix E).

The two oscillations AF1 and AF2 suggest the existence of two different muon stopping sites in the AFM temperature regime. Looking further into the fitting coefficients we can note that AF2 has a substantially larger asymmetry than AF1 ($A_{\mathrm{AF2}} \approx 10A_{\mathrm{AF1}}$). Since asymmetry reflects the fraction of muons experiencing a given local field, the larger asymmetry for AF2 suggests that the majority of muons stopping in the sample do so at the AF2 muon site. Furthermore, the relaxation rate $\lambda_{\mathrm{AF2}}$ is larger than $\lambda_{\mathrm{AF1}}$, indicating a broader field distribution at the site. However, $\lambda_{\mathrm{AF1}}$ has significantly larger error bars, likely due to fitting difficulties from the small volume fraction.

In $MgReO_4$ the AF1 frequency is no longer distinguishable at around $T \geq 60\,\mathrm{K}$. To ensure that the muon site population is truly suppressed rather than simply becoming indistinguishable from the other site as the frequencies decrease, we attempted to fix the ratio between the two frequencies of AF1 and AF2. However, with this, the frequency AF1 is still no longer resolvable at $T = 60\,\mathrm{K}$. This suggests that that one muon stopping site is no longer populated in the range $60\ \mathrm{K} < T < T_N$.

At $T \approx 50\,\mathrm{K}$ we see indications of a small, but sharp jump in asymmetry $A_{\mathrm{AF1}}$, coinciding with a lower precession frequency. While this could be indicative of a second magnetic transition, we suggest that it is more likely an artifact of the fitting, as disregarding the $50\,\mathrm{K}$ point, the frequency evolution seems to follow the power law trend. Furthermore, if there was a magnetic transitions occurring at this temperature, we would expect it to be visible in magnetic susceptibility, which we do not observe [36].

Moreover, as the frequency AF1 disappears, we see a gradual increase in asymmetry for AF2. Our analysis suggests that the disappearance of the AF1 muon site at $60\,\mathrm{K}$ is better explained by changes in the muon site population rather than spin canting, as previously hypothesized in [36](Sec. 3.3).

Finally, we have reanalyzed the critical exponent $\beta$ of $MgReO_4$ by fitting the frequency

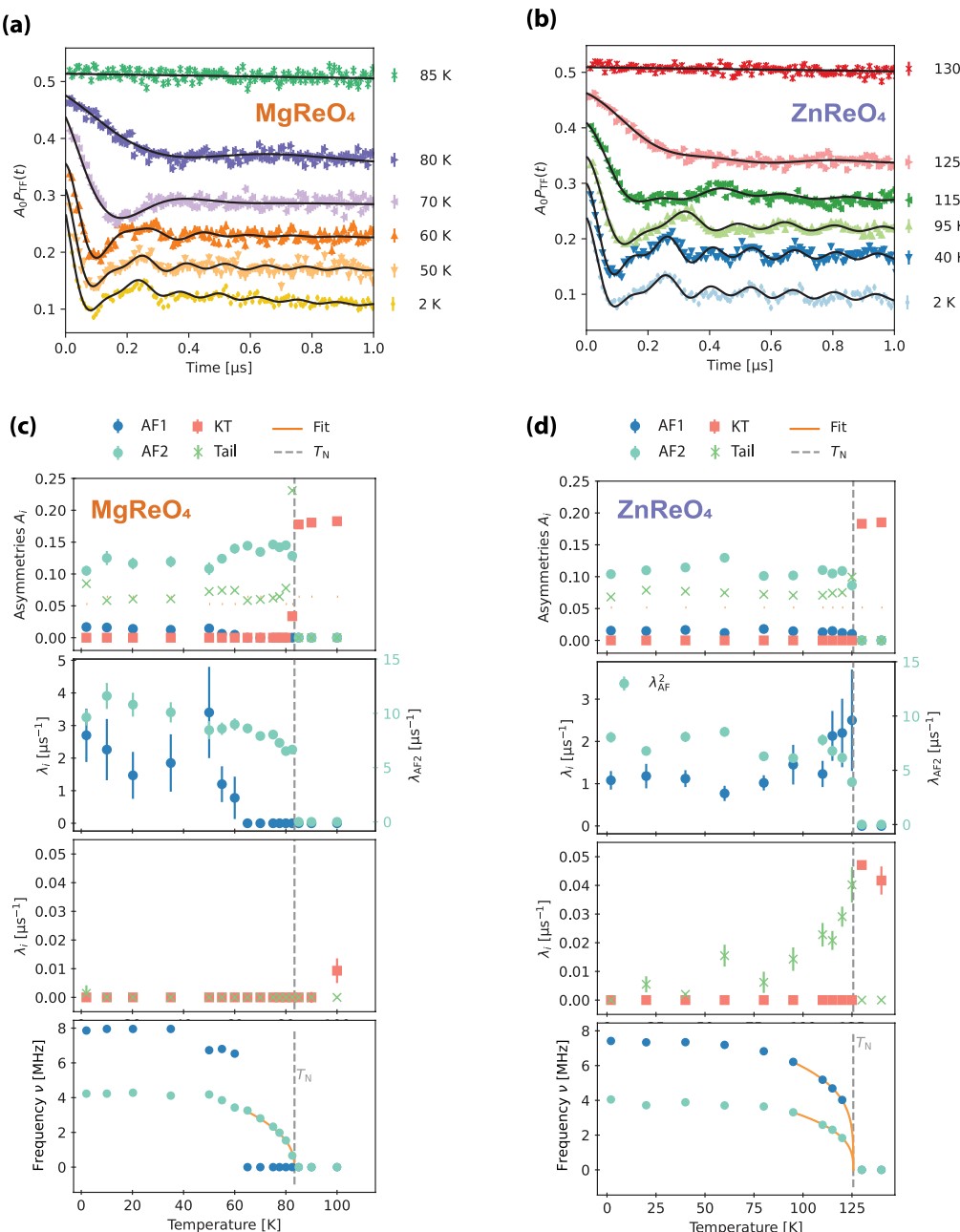

Figure 2: Muon spin spectroscopy ($\mu^+$SR) zero field (ZF) data and corresponding fit lines in black using polarization function (eq. (1)) for **(a)** MgReO$_4$ and **(b)** ZnReO$_4$. Temperature dependent ZF coefficients from polarization function (eq. (1)) for **(c)** MgReO$_4$ and **(d)** ZnReO$_4$. The Néel temperatures, $T_N^{Mg} = 83.35(2)$ K $T_N^{Zn} = 126.67(10)$ K, are marked with a gray line. The oscillation frequencies are fitted with a power law near transition temperature with $\beta_1^{Zn} = 0.35(1)$, $\beta_2^{Zn} = 0.26(1)$ and $\beta_1^{Mg} = 0.41(1)$.

to the mean-field power law:

$$f = \alpha(1 - \frac{T}{T_N})^\beta, \tag{2}$$

using only temperature points near the transition (from $65\,\mathrm{K}$ to a fixed $T_\mathrm{N} = 83.35(2)\,\mathrm{K}$). This fitting yields a critical exponent of $\beta^\mathrm{Mg} = 0.41$, though variations in $T_\mathrm{N}$ within its error margins introduces a $\pm 10\%$ uncertainty, with the choice of temperature range for the fitting contributing an even larger error. The critical exponents is not inconsistent with a three dimensional order parameter [46]. However, the large error due to lack of measurements close to the transition makes it difficult to draw any definitive conclusions about the universality class of the transition.

### 3.2.2   ZnReO$_4$

As in MgReO$_4$, $A_\mathrm{KT}$ dominates above $T_\mathrm{N}$ [Fig. 2 d)], and with lowered temperature this asymmetry decreases in favor of $A_\mathrm{tail}$, $A_\mathrm{AF1}$ and $A_\mathrm{AF2}$, indicative of a magnetic ordering. However, ZnReO$_4$ exhibits a significantly higher $T_\mathrm{N} = 125.67(10)\,\mathrm{K}$ (Appendix E). Similar to MgReO$_4$, it is notable that AF2 has a substantially larger asymmetry ($A_\mathrm{AF2} \approx 7A_\mathrm{AF1}$), again indicating the favoring of muon site AF1. The asymmetry, as an indicator of muon site population, is subsequently used to evaluate the possible magnetic structures of the compounds (Sec. 3.3). Additionally, the relaxation rate $\lambda_\mathrm{AF2}$ is higher than $\lambda_\mathrm{AF1}$ by a factor of two, similar to MgReO$_4$.

We can see that $\lambda_\mathrm{AF1}$, as well as $\lambda_\mathrm{tail}$ reaches a cusp at $T_\mathrm{N}$, as expected near a phase transition. In this case $\lambda_\mathrm{tail}$ is larger and can therefore be fitted more accurately than for MgReO$_4$. In ZnReO$_4$ both frequencies are present throughout the entire AFM phase, indicating that the loss of one observed muon precession frequency is specific to MgReO$_4$

Finally, the critical exponents for ZnReO$_4$, obtained by fitting the frequency from $95\,\mathrm{K}$ to a fixed $T_\mathrm{N} = 126.67\,\mathrm{K}$ using Eq.(2), are $\beta_1^\mathrm{Zn} = 0.35(1)$ and $\beta_2^\mathrm{Zn} = 0.26(1)$ also consistent with a three dimensional order parameter, as in MgReO$_4$. As with MgReO$_4$, the critical exponents have large errors due to the uncertainty in $T_\mathrm{N}$ and the choice of temperature range. The temperature range used is a compromise between the number of data points and staying close to the transition.

## 3.3   Muon site and field calculations

To resolve the magnetic structure and the origin behind the disappearing frequency in MgReO$_4$, we conducted self consistent calculations using Quantum ESPRESSO [40] to determine the electrostatic potentials in the compounds. Here, we did not consider local distortions due to the implanted muon, as the two compounds do not contain mobile ions and their structures are already quite distorted. Therefore, any small perturbation from the muon is unlikely to have a significant effect. Our calculations suggest two potential muon sites [Tab. 2], which are only slightly shifted along the $b$ axis between the compounds.

First, to confirm the muon site calculations, we calculate the spherical average of the internal field distribution width arising from the nuclear moments, $\Delta$ as

$$\Delta_\mathrm{ZF}^2 = 2 \left( \frac{\mu_0}{4\pi} \right)^2 \sum_i \frac{\gamma_i^2 \hbar^2}{r_i^6} \frac{I_i(I_i + 1)}{3}, \tag{3}$$

where $r_i$ is the distance of the $i^\mathrm{th}$ nucleus, and $\gamma_i$ and $I_i$ are the gyromagnetic ratio and nuclear spin of the nucleus [47]. The calculated values [Tab. 2] fit the experimental values quite well.

With the muon sites confirmed, we calculated local magnetic fields for various spin structures based on our previous approaches [48–50]. We assume the local field is primarily dipolar, as hyperfine coupling effects have been shown to be minimal, even in an A-type AFM [48]. To calculate the possible magnetic structures, we use irreducible representation

(IR) analysis, a powerful symmetry-based method that places strict limits on the allowed moment directions for a given crystal structure [51].

From the literature on other magnetic wolframite compounds, $FeWO_4$ is reported to adopt an AFM structure with $\mathbf{k} = (1/2, 0, 0)$ and moments along $a$, while under hydrostatic pressure, the spins become canted in the $ac$-plane [52]. Similarly, $NiWO_4$ [53] and $MnWO_4$ [54] exhibit canted AFM spin structures in the $ac$-plane. In these compounds, the magnetic ions occupy the $2f$ Wyckoff site, while Re in our compounds is at the $2e$ site. This difference swaps the allowed moment directions for each IR: those permitting moments in the $ac$-plane for the $2f$ site correspond to moments only allowed along $b$ in our case, and vice versa.

Based on the related wolframite compounds, we calculated the local magnetic field for $\mathbf{k} = (0, 0, 0)$, $\mathbf{k} = (0, 1/2, 0)$ and $\mathbf{k} = (1/2, 0, 0)$. When calculating the different spin configurations, we accounted for the probability distribution of the muon sites and used that the size of the measured asymmetry is proportional to the muon stopping probability. The three $\mathbf{k}$-vectors, $\mathbf{k} = (0, 0, 0), (0, 1/2, 0), (1/2, 0, 0)$ share the same 4 IRs [Table 3], two of which allow the spins only along the $b$-axis, while the other two allow spins to point in the $ac$-plane, but depending on the $\mathbf{k}$-vector they result in different magnetic structures. For $\mathbf{k} = (0, 0, 0)$, $\Gamma_1$ and $\Gamma_3$ are FM structures, while $\Gamma_2$ and $\Gamma_4$ are AFM. For $\mathbf{k} = (0, 1/2, 0)$ and $(0, 0, 1/2)$, all structures are AFM, with $\Gamma_4$ yielding the same structure as for $\mathbf{k} = (0, 0, 0)$.

For completeness, the local field based on a magnetic structure with $\mathbf{k} = (0, 0, 1/2)$ was also calculated. This wave vector has only one IR, which, when canted, forms a non-collinear structure. This structure is tunable through internal degrees of freedom and can also result in a collinear configuration with spins along $b$. However, this structure was not consistent with the measured frequencies in ZF $\mu^+$SR.

The other $\mathbf{k}$-vectors revealed three possible spin structures which we found could reproduce the measured $\mu^+$SR frequencies. One structure, $\Gamma_3$ with $\mathbf{k} = (1/2, 0, 0)$ has a magnetic moment of $\mu = 0.77\mu_B$, because the muon sites are positioned between two AFM coupled spins. As a result, a larger magnetic moment is needed to induce the same muon precession frequencies. This structure was disregarded, as such a large moment would be detectable with NPD. The remaining two structures, which both involve spin canting away from the principal axes in the $ac$-plane and exhibit low magnetic moments, closely match the experimental data [Tab. 2]. The difference between the structures is that in $\Gamma_3$ with $\mathbf{k} = (0, 1/2, 0)$, the Re atoms that are further separated along $b$ are FM coupled, while in $\Gamma_4$ with $\mathbf{k} = (0, 0, 0)$, they are AFM coupled [Fig.3].

To explore the possibility of a reduction in the number of precession frequencies due to spin canting in $MgReO_4$, we allow the spins to rotate in the $ac$-plane, which is the only distortion direction allowed for the $\Gamma_3$ and $\Gamma_4$ IRs. Our calculations show that the two muon sites can begin to experience the same internal magnetic field at specific canting angles for $\Gamma_4$. However, this would mean that the lower frequency AF2 would rise to 6 MHz, which we do not observe. Otherwise, the frequency ratio could be consistent with canting if both magnetic sites experienced a sudden drop in the magnitude of the moment, which we would not expect at a spin canting transitions when there is no accompanying structural transition (see Appendix F). Therefore, it is unlikely that spin canting is responsible for the disappearing frequency.

## 3.4  Bond Valence Sum

The magnetic structure and moment size of $MgReO_4$ and $ZnReO_4$ are very similar. Interestingly, $Re^{6+}$ has been previously reported to have an effective moment of $\mu = 1.2$ - $1.7\mu_B$, while $Re^{5+}$ and $Re^{7+}$ have been reported to have much weaker effective moment of

| MgReO$_4$ | | | |
|---|---|---|---|
| Muon site | $f_{\text{AFM}}$ (MHz) | $\Delta_{\text{MgReO}_4}$ ($\mu$s$^{-1}$) | $\Delta_{\text{KT}}$ ($\mu$s$^{-1}$) |
| $\mu_1$(0.0 0.37 0.75) | 4.229(9) | 0.135 | 0.232(11) |
| $\mu_2$(0.5, 0.12, 0.25) | 7.86(7) | 0.156 | 0.232(11) |

| Γ3 | | |
|---|---|---|
| $k = (0, \frac{1}{2}, 0)$ | $\mu = 0.295\mu_{\text{B}}$ | Canting angle 51° |
| Muon site | $B_{\text{dip}'}(G)$ | $f_{\text{dip}'}$ (MHz) |
| $\mu_1$ | [ 0.01306, 0, -0.02835] | 4.23 |
| $\mu_2$ | [ 0.04902, 0, -0.03136] | 7.89 |

| Γ4 | | |
|---|---|---|
| $k = (0, 0, 0)$ | $\mu = 0.255\mu_{\text{B}}$ | Canting angle 37.5° |
| Muon site | $B_{\text{dip}'}(G)$ | $f_{\text{dip}'}$ (MHz) |
| $\mu_1$ | [-0.05291, 0, 0.02301] | 4.185 |
| $\mu_2$ | [0.00097, 0, -0.03086] | 7.82 |

| ZnReO$_4$ | | | |
|---|---|---|---|
| Muon site | $f_{\text{AFM}}$ (MHz) | $\Delta_{\text{ZnReO}_4}$ ($\mu$s$^{-1}$) | $\Delta_{\text{KT}}$ ($\mu$s$^{-1}$) |
| $\mu_1$(0.0 0.39 0.75) | 4.05(5) | 0.136 | 0.1207(4) |
| $\mu_2$(0.5, 0.14, 0.25) | 7.42(3) | 0.158 | 0.1207(4) |

| $\Gamma_3$ | | |
|---|---|---|
| $k = (0, \frac{1}{2}, 0)$ | $\mu = 0.292\mu_{\text{B}}$ | Canting angle 56° |
| Muon site | $\mathbf{B}_{\text{dip}'}(G)$ | $f_{\text{dip}'}$ (MHz) |
| $\mu_1$ | [-0.01209, 0, 0.02725] | 4.04 |
| $\mu_2$ | [ 0.04397, 0, -0.03278] | 7.43 |

| $\Gamma_4$ | | |
|---|---|---|
| $k = (0, 0, 0)$ | $\mu = 0.242\mu_{\text{B}}$ | Canting angle 39.5° |
| Muon site | $\mathbf{B}_{\text{dip}'}(G)$ | $f_{\text{dip}'}$ (MHz) |
| $\mu_1$ | [-0.00027 0. 0.02994] | 4.06 |
| $\mu_2$ | [ 0.04981 0. -0.02268] | 7.42 |

Table 2: The obtained muon sites and experimental data for MgReO4 and ZnReO4 are presented, along with calculated dipolar fields for two proposed magnetic structures, $\Gamma_3$ and $\Gamma_4$. The table includes the internal field distribution of nuclear moments ($\Delta$KT), experimentally obtained frequencies ($f$AFM from ZF fit), and field distributions ($\Delta_{\text{MgReO}_4}$ and $\Delta$ZnReO4 from fitting). Also shown are the magnetic structures, wave vector $k$, moment $\mu$, and canting angle (relative to the $a$-axis). Muon precession frequencies are calculated as $f\text{dip}' = (\gamma_\mu/2\pi)|\mathbf{B}\text{dip}'|$, where $\mathbf{B}\text{dip}'$ is the dipolar magnetic field at the muon site and $\gamma_\mu$ is the muon gyromagnetic ratio.

$\mu = 0.3$ - $0.8\mu_{\text{B}}$ [20]. Considering the expected value of the oxidation of Re$^{6+}$ being much higher than our estimation, as well as not visible in NPD, it may be appropriate to assign different oxidation states for Re in MgReO$_4$ and ZnReO$_4$. We therefore employ bond

Table 3: Irreducible representations (IRs) of the little group for propagation vectors $\mathbf{k} = (0,0,0), (0,1/2,0), (1/2,0,0)$ in space group $P2/c$. The symmetry operators are given in Seitz notation and the IRs are calculated using BasIreps. The allowed moment directions are indicated for each IR.

| IR | $\{1\|000\}$ | $\{2_{0y0}\|00p\}$ | $\{-1\|000\}$ | $\{m_{x0z}\|00p\}$ | Moment |
|----|------|------|------|------|------|
| $\Gamma_1$ | 1 | 1 | 1 | 1 | (0,v,0) |
| $\Gamma_2$ | 1 | 1 | -1 | -1 | (0,v,0) |
| $\Gamma_3$ | 1 | -1 | 1 | -1 | (u,0,w) |
| $\Gamma_4$ | 1 | -1 | -1 | 1 | (u,0,w) |

valance sum (BVS) analysis which uses the bond lengths to nearby atoms to estimate the oxidation state of the atom. BVS analysis assumes an perfect correlation between bond length and oxidation state, which may be influenced by local structural distortions. Further validation through complementary techniques, such as X-ray absorption spectroscopy (XAS), could strengthen these findings.

The bond length between atoms is directly correlated with the oxidation state, generally resulting in a higher oxidation state when the bond length is shorter. The bond length is however also dependent on the atom size, which is not a well defined quantity, and depends on the charge carrier density around the atom. There are multiple ways to calculate the relation between bond length and bond flux and one of the most simple and still very robust methods is BVS. For more details of the method see Ref. [56].

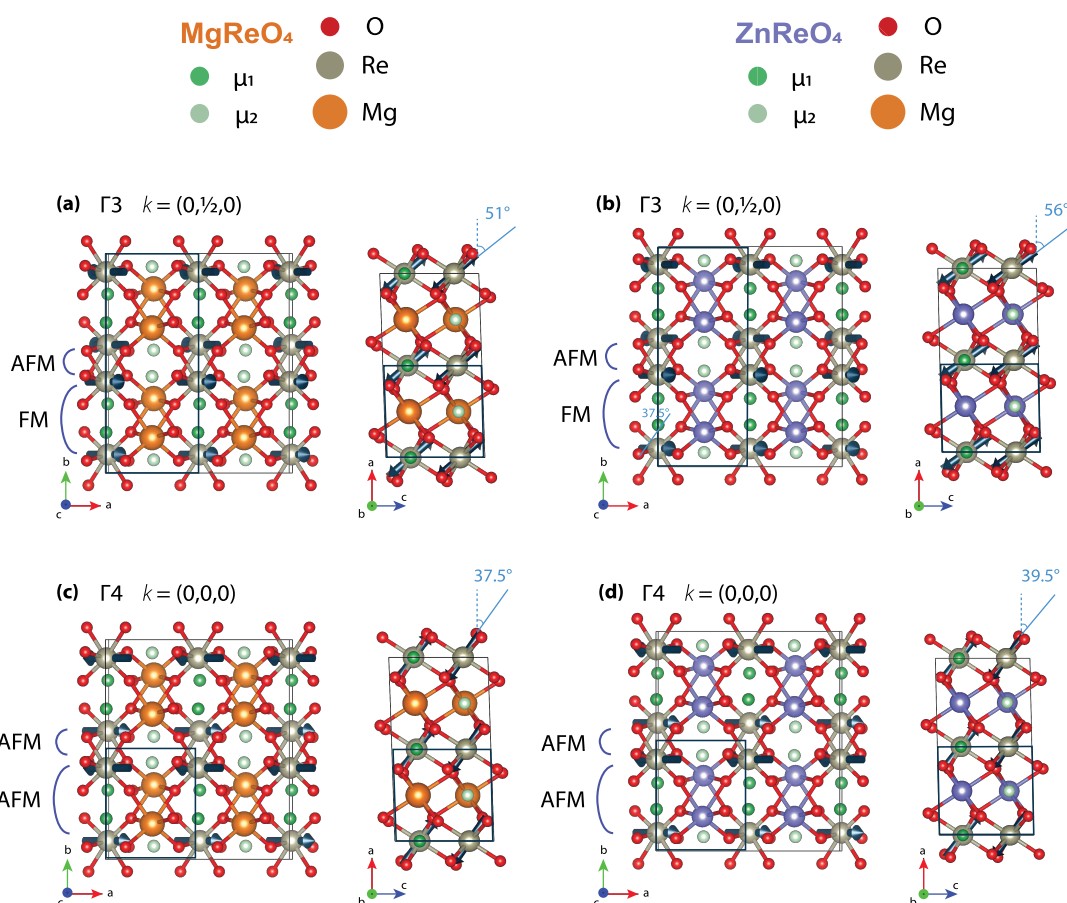

Figure 3: Proposed magnetic structures for irreducible representations (IRs) $\Gamma_3$ and $\Gamma_4$, drawn with VESTA [55]. **(a, b)** $\Gamma_3$ for MgReO$_4$ and ZnReO$_4$, respectively. **(c, d)** $\Gamma_4$ for MgReO$_4$ and ZnReO$_4$. The unit cell is doubled along $a$ and $b$, with the magnetic unit cell outlined in blue. Muon sites (green) and spin structures (dark arrows) are shown. In both cases, the moments lie in the $ac$-plane, canting away from the $a$-axis by different amounts corresponding to the values which reproduce the experimental frequencies found in $\mu^+$SR.

In BVS, the valence $V$ is defined as as

$$V = \sum_i s_i = \sum_i e^{(R_0 - R_i)/b}, \tag{4}$$

where $R_0$ is the reference bond length corresponding to valence $+1$, specific to each cation-anion pair, $R_i$ is the distance between atom $i$ and the center atom and $s_i$ is the experimentally obtained bond valence. The parameter $b$ controls how rapidly the bond valence decreases or increases with a change in bond length. For most oxides, $b$ is empirically found to be approximately 0.37 Å [57].

Since our system involves typical ionic or covalent bonds, the standard value of $b = 0.37$ Å is appropriate. Since $R_0$ varies between each kind of pair of atoms it needs to first be empirically determined from a set of well characterized compounds. However, the empirically determined $R_0$ can only be applied for atoms having a similar range of bond lengths. An extensive list of bond valences has been summarized by Brown [58]. Unfortunately,

no value for $R_0$ is reported for $Re^{6+}$. However, for $Re^{7+}$, a value of $R_0^{Re7+} = 1.943$ Å is given [59], and unconfirmed reports for other oxidation states suggest values around $R_0 \approx 1.9$ Å [58]. Given the similar atomic sizes, it is likely that $R_0$ for $Re^{6+}$ does not differ significantly from this value.

To find the value of $R_0^{Re6+}$, we have looked at other $Re^{6+}$ compounds ($MnReO_4$ [35] and $Sr_3Re_2O_9$ [26]) BVS. We begin by calculating the BVS for Mn and Sr in the reference materials, both of which are close to the expected value of 2 [Tab. 4]. Then, we determine $R_0$ for $Re^{6+}$ by fitting it using a least-squares approach and known crystal structures. The obtained $R_0$ values are very similar, ranging from 1.91 to 1.92. Since this is close to the reported $R_0$ value for $Re^{7+}$, it gives us confidence that the BVS values should be reliable for our materials.

| Compound | Bond | $V$ | $R_0$ [Å] |
|---|---|---|---|
| $MnReO_4$ | Mn - O | 1.948 | 1.740 [60] |
| $Sr_3Re_2O_9$ | Sr - O | 2.084 | 1.765 [61] |
| | | | |
| Fitted | $MnReO_4$ | $Sr_3Re_2O_9$ | |
| $R_0^{6+}$ | 1.9111(7) | 1.9206(5) | |

Table 4: Bond valence sums (BVS) for Mn-O in $MnReO_4$ (structure [35]) and Sr-O in $Sr_3Re_2O_9$ (structure [26]), along with the fitted $R_0$ values for Re-O for these compounds.

| $MgReO_4$ | Bond | $V$ | $R_0$ [Å] |
|---|---|---|---|
| | Mg - O | 1.981 | 1.693 [62] |
| | Re - O | 6.091 | 1.9111(7) |

| $ZnReO_4$ | Bond | $V$ | $R_0$ [Å] |
|---|---|---|---|
| | Zn - O | 1.903 | 1.704 [62] |
| | Re - O | 6.55 | 1.9111(7) |

Table 5: Calculated bond valence sums (BVS) for $MgReO_4$ and $ZnReO_4$, using $R_0 = 1.91$ from $MnReO_4$ for Re - O (Tab. 4).

We can now use this to investigate the BVS in $MgReO_4$ and $ZnReO_4$ (Tab. 5). To apply the BVS analysis, the bond lengths of the reference sample must be comparable. We chose the $R_0$ value for $Re^{6+}$ from $MnReO_4$, as it exhibits the most similar bond length variation to our samples (1.80 Å - 2.05 Å). The bond length ranges in our samples are 1.79 Å to 2.08 Å for $MgReO_4$ and 1.74 Å to 2.10 Å for $ZnReO_4$.

Unfortunately, the bond length range for $ZnReO_4$ is more than 0.1 Å larger than that for the reference sample $MnReO_4$, making it difficult to reliably calculate the BVS for $ZnReO_4$. This larger variance may stem from Zn having a full $3d$ electron shell, which leads to stronger ligand interactions and increased flexibility in bond lengths. Additionally, $Zn^{2+}$ is susceptible to second-order JT distortions, causing asymmetries in the ligand arrangement [63]. In contrast, $MgReO_4$ does not experience these effects as strongly because $Mg^{2+}$ lacks $d$-electrons, resulting in weaker ligand-metal interactions and a more symmetric structure.

For $MgReO_4$ our calculations results in BVS, $V_{Re-O} = 6.115$ and $V_{Mg-O} = 1.981$, suggesting $Re^{6+}$. Since $MgReO_4$ and $ZnReO_4$ exhibit similar magnetic behavior as seen from $\mu^+SR$, we can reasonably assume the Re to be in the same oxidation state $Re^{6+}$ for

$ZnReO_4$ as well.

# 4 Discussion

Both compounds have a wolframite-type nuclear structure with distorted octahedral coordination of O around the Re atoms. It is in fact not uncommon for $Re^{7+}$, $Re^{6+}$ and $Re^{5+}$ to stabilize in distorted octahedron configuration with O [20, 64]. In other compounds, distortions of the octahedral center has been shown to be due to Re - Re bonds within the oxide, causing varying distances between the Re atoms [33, 65]. In our study, the observed Re-Re distances are relatively long - about 4.5 Å for both $MgReO_4$ and $ZnReO_4$ - compared to the bond length in $ReO_2$, which is 2.48 Å [33], making Re-Re bonds unlikely to be responsible for the distortions.

A possible cause of the distorted octahedra is JT distortions, which occur in octahedral complexes with degeneracy in the $d$-orbitals [66]. To break the degeneracy, the structure undergoes distortion, lowering the crystal symmetry. This type of distortion is plausible for $5d^1$ electron systems, such as $Re^{6+}$, and could therefore cause the distorted octahedral configuration in our compounds. Second-order JT distortions from $Zn^{2+}$ could account for the more distorted structure found in $ZnReO_4$.

The ZF polarization function coefficients of the two compounds hold many similarities [Fig. 2]. At low temperatures, both compounds display two oscillations, AF1 and AF2, with similar frequencies (about 4 MHz and 8 MHz), indicating the presence of two muon stopping sites. The relaxation rates, $\lambda_i$, are also similar. A difference between the compounds is that for $ZnReO_4$, $\lambda_{\text{tail}}$ reaches a cusp at $T_N$, as expected with the onset of fluctuations at a phase transition. This kind of cusp is however not visible for $MgReO_4$, which could be due to the even smaller asymmetry of AF1 in $MgReO_4$ or fitting difficulties due to the impurity fraction.

The main distinction between the compounds is the number of frequencies reduces to one in $MgReO_4$ for $T \geq 60$ K, which we previously speculated to be due to spin canting [36]. Here, we suggest that spin canting is unlikely to be the cause of the missing frequency. Instead, we note that the volume fraction of AF1 is very small, and uncertainty in $\lambda_{\text{AF1}}$ high. It is therefore possible that small perturbations, such as thermal expansion of the unit cell or changes in the octahedral distortion, could alter the electric field distribution in such a way that the muon site is no longer populated. A temperature dependent structural study, combined with corresponding DFT calculations of the electronic structure could resolve this scenario.

The two suggested magnetic structures [Fig. 3] have low magnetic moments ($0.29(5)\mu_B$ for $\Gamma_3$ and $0.25(8)\mu_B$ for $\Gamma_4$), much lower than what is expected for $Re^{6+}$ (1.2 - 1.7 $\mu_B$). The low moment is further confirmed by lack of a magnetic contribution detectable in the NPD patterns. BVS calculations confirm $MgReO_4$ to be in the $Re^{6+}$ oxidation state, with likely a similar result for $ZnReO_4$. This raises the question of why the magnetic moment is so low in both $MgReO_4$ and $ZnReO_4$, which $\mu^+SR$ data shows is of similar magnitude.

In fact other $Re^{6+}$ oxides have been found exhibiting moments as low as 0.3 - 0.8 $\mu_B$ [43, 67, 68]. This suppression is attributed to SOC being strong enough to cause a splitting in the $t_{2g}$ orbitals into a lower energy $J_{eff} = 3/2$ quartet and higher energy $J_{eff} = 1/2$ doublet [69]. The quartet has a theoretical effective magnetic moment of zero, but a non-zero moment is typically observed. In a DFT study addressing $5d^1$ electron systems, such as $Re^{6+}$, this effect has been attributed to hybridization between the $d$ orbital with the ligand $p$ orbitals [70]. In the same study, they also show that distortion to the octahedral configuration also causes an increase in magnetic moment, but that this quite quickly saturates as it distorts from an ideal octahedra, aligning with the fact that we see similar magnetic moment in both compounds, even when $ZnReO_4$ has a more distorted octahedral configuration.

We investigated two compounds with remarkably similar magnetic properties at base temperature, despite Zn and Mg having different electronic configurations. This suggests that spin-orbit coupling (SOC) is responsible for the suppressed moment, consistent with similar effects observed in other $Re^{6+}$ oxides. Bramnik $et$ $al.$ [35] also observed a lower-than-expected paramagnetic moment in $MnReO_4$, considering a different oxidation state scenario ($Mn^{2+}/Re^{7+}$), but ruled this out based on bond lengths. Their analysis similarly supports that the compound remains in the $Re^{6+}$. This indicates that the suppressed magnetic moment is a characteristic feature of $Re^{6+}$ compounds with octahedral oxygen coordination, largely independent of the other metals in the material.

Further, studying the temperature dependence of the lattice parameters, specifically across $T_N$, would deepen our understanding of the relationship between octahedral distortion and magnetic moment in these $Re^{6+}$ oxides.

# 5 Conclusions

Using NPD we have been able to characterize the monoclinic wolframite structure of $ZnReO_4$ and $MgReO_4$, completing the structure description first reported by Sleigh $et$ $al.$ [23]. Both compounds show distorted octahedral coordination of O around the Re, with larger distortion in $ZnReO_4$.

Using $\mu^+SR$ and ACMS, we have found $MgReO_4$ and $ZnReO_4$ to be AFM with respective $T_N = 83.35(2)$ and $T_N = 125.67(10)$. Furthermore, in the AFM phase, we find two muon sites with a similar temperature dependence in $ZnReO_4$, in contrast to $MgReO_4$, where the number of frequencies reduces to one at $T \geq 60\,K$. Our results are most consistent with one muon site becoming suppressed (less populated) with higher temperature, which could be caused by a temperature-induced change in the electronic structure that significantly decreases the muon stopping probability for that site. High-resolution structural measurements and DFT calculations would help clarify this.

Muon site calculations reveal two possible AFM structures, with the spins canted away from the $a$-axis in the $ac$-plane: $\Gamma_3$ with $\mathbf{k} = (0, 1/2, 0)$ and $\Gamma_4$ with $\mathbf{k} = (0, 0, 0)$, for both $MgReO_4$ and $ZnReO_4$. The ordered moments of $0.29(5)\mu_B$ for $\Gamma_3$ and $0.25(8)\mu_B$ for $\Gamma_4$ are consistent with the lack of magnetic contribution seen in the NPD patterns below $T_N$. In this case, the muon, being a highly sensitive magnetic probe, is one of the only ways of

getting insight into the subtle magnetic properties of these materials.

The obtained ordered magnetic moment is significantly smaller than the expected moment for mononuclear $Re^{6+}$ compounds (1.2 - 1.7 $\mu_B$). Our bond valence sum (BVS) analysis shows that the $Re^{6+}$ state applies for $MgReO_4$ and most likely also applies for $ZnReO_4$. Therefore, we propose that SOC together with $d$ - $p$ ligand hybridization is the cause of the suppressed observed ordered moment.

We investigated two compounds with similar magnetic properties, despite different electronic configurations in Zn and Mg. Our results suggest that SOC suppresses the magnetic moment, a feature consistent with other $Re^{6+}$ oxides. This suppressed moment appears to be characteristic of $Re^{6+}$ compounds with octahedral oxygen coordination, regardless of the other metals present in the compound.

## Acknowledgements

We would like to thank A. Kentaro Inge from Stockholm University for his assistance with the XRD. NPD beamtime was performed at J-PARC using the iMATERIA instrument (Proposal Number: 2024A0387), and we are grateful for the technical assistance provided by the local staff. $\mu^+$SR measurements were conducted at the General Purpose Surface Muon (GPS) instrument at PSI (Experimental number: 20221283). We sincerely appreciate the invaluable support and expertise of the facility staff.

**Funding information**   The research is funded by the Swedish Research Council, VR (Dnr. 2021-06157 and Dnr. 2022-03936), the Swedish Foundation for Strategic Research (SSF) within the Swedish national graduate school in neutron scattering (SwedNess), the Carl Tryggers Foundation for Scientific Research (CTS-22:2374), and the Knut and Alice Wallenberg Foundation through the grant 2021.0150. U.M. acknowledges funding from the KTH-SCI doctoral excellence program. J.S. was supported by the Japan Society for the Promotion of Science (JSPS) KAKENHI Grant No. JP23H01840 and JP24H00042. O.K.F. is supported by the Swedish Research Council (VR) via a Grant 2022-06217 and the Foundation Blanceflor 2023 and 2024 fellow scholarships. E.N. acknowledges financial support from the SSF-Swedness grant SNP21-0004 and the Foundation Blanceflor 2024 fellow scholarship.

**Data Availability and Analysis:**   The data and analysis supporting this study are available from the corresponding authors upon reasonable request.

## A   AC Magnetic Susceptibility

The real part of the AC magnetic susceptibility (ACMS) signal as a function of temperature can be found in Fig. 4. From the measurement, we can identify $T_N \approx 126$ K. The sharp increase below $T = 100$ K is likely due to ferromagnetic (FM) impurity, similar as seen in transportation measurements for $MgReO_4$ [36] and which can be identified as a paramagnetic background in the muon spin spectroscopy ($\mu^+$SR) data. As the cusp is very small, we see that the magnetic signal is quite weak. Due to the impurities, it is difficult to fit the ACMS data to extract the magnetic moment. We should also note that some more sample degradation may have occurred in the sample mounting process as the capsule was exposed to air during transport.

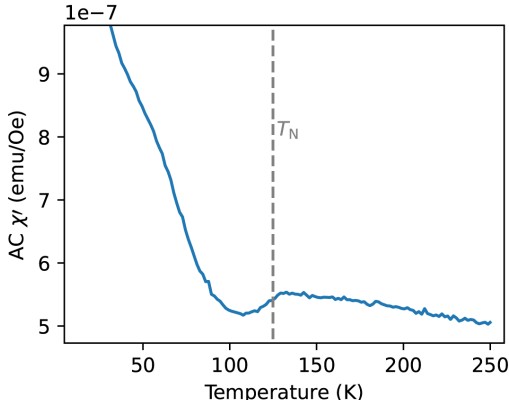

Figure 4: Real part of AC Magnetic Susceptibility signal of $ZnReO_4$ as a function of temperature. Grey line indicates $T_N = 125.67\,\text{K}$ obtained from $\mu^+SR$

# B   XRD analysis

In an attempt to determine the crystal structure and the atomic positions in $MgReO_4$ and $ZnReO_4$, XRD patterns were collected at room temperature. The patterns were collected using the Single Crystal X-ray Diffractometer Bruker D8 VENTURE (1:2 ratio of $CuK\alpha_1$ and $CuK\alpha_2$ radiation, Göbel mirror, transmission mode, step size $0.02°$). Because of the high X-ray absorption of rhenium, the sample was mounted on a glass fiber coated in Dow Corning high vacuum grease. The glass fiber was then mounted in a quartz capillary (diameter $0.3\,\text{mm}$) and sealed in an argon environment. The diffraction spectra were refined using `FULLPROF SUITE` [38]. We collected approximately double the statistics in the XRD pattern for $MgReO_4$ than $ZnReO_4$.

After refinement of the neutron powder diffraction (NPD) data, it was understood that the samples must have began degrading by reacting with the hydrogen present in the grease in the mounting procedure. The refined structures are highly distorted, especially for $ZnReO_4$. Following is the description of the refinement from the XRD measurements.

The measured peaks [Fig. 5 a)-b)] could best be identified using a monoclinic wolframite structure with the space group $P2/c$ (# 13) [44]. The refined lattice parameters [Tab. 6] are close to the values found by Sleigh *et al.* [23]. The structure [Fig. 5 c)-d)] contains edge-sharing distorted octahedral coordination of O surrounding the Re atoms. The octahedra form a zig-zag formation along the $c$-axis and sandwiches the Mg/Zn atoms along the $a$-axis [Fig.5 c)-d)]. In the distorted octahedra of $MgReO_4$, the Re-O-Re bond angles range from $73°$ to $111°$, while in $ZnReO_4$, they vary from $62°$ to $136°$, both significantly deviating from the ideal $90°$. Additionally, the octahedra are compressed along the $a$-axis and elongated along the $c$-axis. The Re center is also displaced along the $b$-axis, leading to varying Re-Re distances along this direction.

Both samples exhibit large impurity peaks around $2\theta = 25° - 30°$. The impurities could not be refined to any known compound containing O, Re and/or Mg/Zn and the regions containing these peaks were therefore excluded in the refinement.

| Parameter | ZnReO$_4$ |
|---|---|
| Crystal structure | monoclinic P2/$c$ |
| $a, b, c$ (Å) | 4.69510, 5.60900, 5.02330 |
| $\beta$ (°) | 91.2600 |
| $V$ (Å$^3$) | 132.2557 |
| Zn $(x, y, z)$ | (0.5, 0.67400, 0.25) |
| Re $(x, y, z)$ | (0, 0.16850, 0.25) |
| O1 $(x, y, z)$ | (0.17100, 0.10500,1.02600) |
| O2 $(x, y, z)$ | (-0.05500, 0.66100, 0.42700) |
| | |
| Refinement | |
| $\chi^2$ | 24.49 |
| Excluded region (°) | 0 - 14, 25.3 - 30.0 |

| Parameter | MgReO$_4$ |
|---|---|
| Crystal structure | monoclinic P2/$c$ |
| $a, b, c$ (Å) | 4.68210, 5.57500, 5.00710 |
| $\beta$ (°) | 92.0260 |
| $V$ (Å$^3$) | 130.619952 |
| Mg $(x, y, z)$ | (0.5, 0.0.67600, 0.25) |
| Re $(x, y, z)$ | (0, 0.16330, 0.25) |
| O1 $(x, y, z)$ | ( 0.11900, 0.16200, 0.91200) |
| O2 $(x, y, z)$ | ( -0.19800, 0.60500, 0.60200) |
| | |
| Refinement | |
| $\chi^2$ | 12.16 |
| Excluded region (°) | 0 - 14, 25.5 - 28.0 |

Table 6: Crystal structure parameters of ZnReO$_4$ and MgReO$_4$ from refinement of powder XRD data.

# C   Zero Field

We can verify the origin of the tail component in the zero field (ZF) polarization function by checking that the sum of AFM asymmetries ($\frac{2}{3}$ of the signal) divided by two corresponds to the tail signal. Fig. 6 shows that these quantities align well in our case below $T_N$ for ZnReO$_4$, implying that the tail component indeed arises due to spins parallel to the initial muon spin polarization.

# D   Longitudinal Field

Longitudinal field (LF) measurements were conducted in order to confirm a suitable ZF polarization function to be used above $T_N$. The moments are decoupled already at $20\,\text{G}$ in both MgReO$_4$ and ZnReO$_4$, indicating static nuclear spin distribution above $T_N$. The most suitable polarization function is therefore KT described by the following polarization function

$$A_0 P_{\text{KT}}(t) = A_{\text{KT}} G^{\text{SGKT}}(t, \Delta_{\text{KT}}) e^{-\lambda_{\text{KT}} t} \tag{5}$$

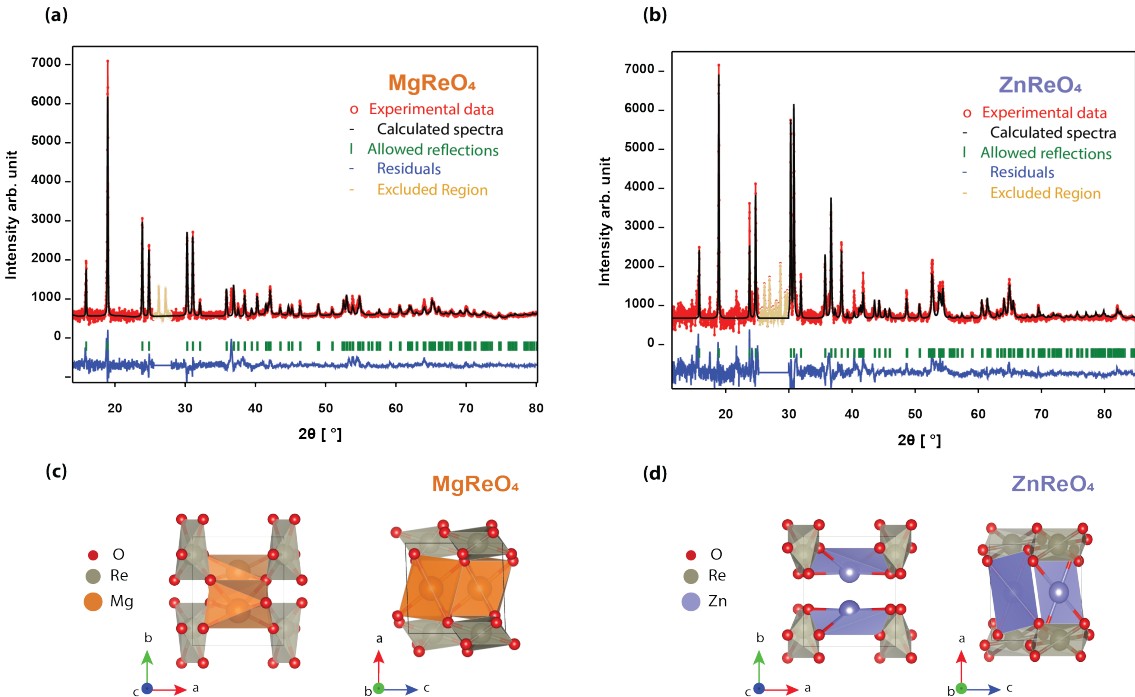

Figure 5: Powder X-ray diffraction pattern for **(a)** ZnReO$_4$ and **(b)** MgReO$_4$ respectively, showing the measured pattern, fitted spectra, allowed reflections and residuals. Excluded regions, due to impurities are removed in the plot. Monoclinic wolframite structure of **(c)** ZnReO$_4$ and **(d)** MgReO$_4$, showing the highly distorted oxygen octahedra formed around Re atoms in the $a - b$ plane and $a - c$ plane due to sample degradation.

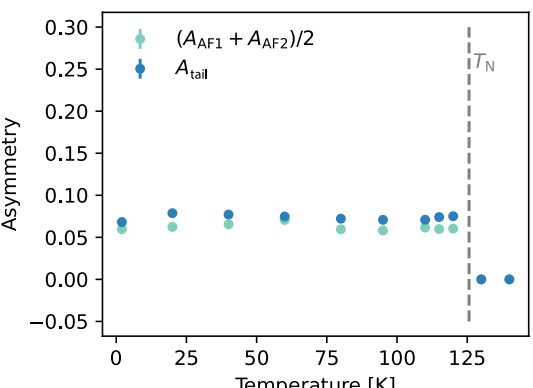

Figure 6: The sum of asymmetries of the two AF oscillatory terms from polarization function (eq. (1)) in ZnReO$_4$ divided by two plotted together with the tail asymmetry.

where $A_{\mathrm{KT}}$ is the KT asymmetry, $\lambda_{\mathrm{KT}}$ is the exponential damping, and $\Delta_{\mathrm{KT}}$ the Gaussian distribution width of the KT. Appendix Fig. 7 shows the measured spectra and fitted polarization function above $T_{\mathrm{N}}$ for MgReO$_4$ and ZnReO$_4$ for an applied LF of $0\,\mathrm{G}$, $10\,\mathrm{G}$ and $20\,\mathrm{G}$. Here, the total asymmetry and field distribution $\Delta_{\mathrm{KT}}$ is common for all applied fields.

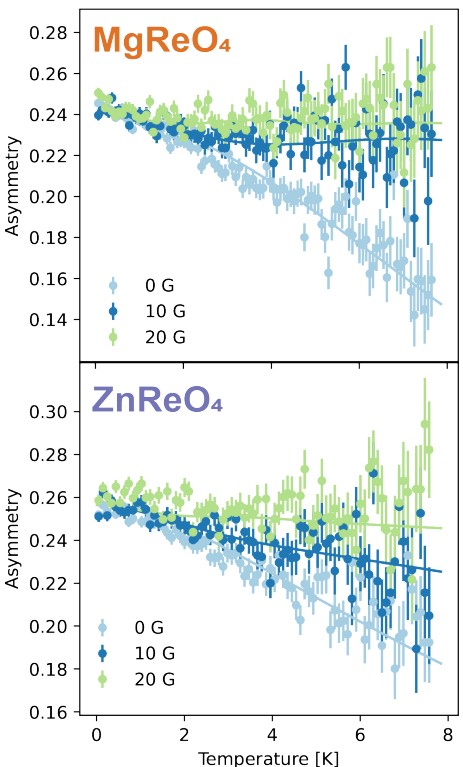

Figure 7: Longitudinal field (LF) measurements and fitted Kubo-Toyabe (KT) polarization function of MgReO$_4$ and ZnReO$_4$ for applied fields of 0 G, 10 G and 20 G at temperatures of 100 K and 140 K respectively.

# E   Transverse Field

$\mu^+$SR measurements in applied transverse field (TF) of $B = 50$ G for both MgReO$_4$ and ZnReO$_4$ (selected temperatures for ZnReO$_4$ visible in Fig. The data [Fig. 8]) is best described by the polarization function:

$$
\begin{aligned}
A_0 P_{\text{TF}}(t) =& A_{\text{TF}} \cos\left(2\pi f_{\text{TF}} + \frac{\pi \phi_{\text{TF}}}{180}\right) e^{-\lambda_{\text{TF}} t} \\
& + A_{\text{AF}} \cos\left(2\pi f_{\text{AF}} + \frac{\pi \phi_{\text{AF}}}{180}\right) e^{-\lambda_{\text{AF}} t} \\
& + A_{\text{tail}} e^{-\lambda_{\text{tail}} t},
\end{aligned}
\tag{6}
$$

where $P_{\text{TF}}(t)$ is the muon polarization in TF and $A_0$ the total asymmetry, which for our experimental setup is $A_0 \approx 0.25$. The polarization function is made up of three contributions; two relaxed oscillating terms coming from the applied TF and inner AFM field, and a slowly decaying magnetic tail arising from the internal magnetic moments which are parallel to the muon spin. In the polarization function, $A_i$ denotes the corresponding asymmetry, $\lambda_i$ the damping, and $f_i$ and $\phi_i$ the frequency and phase of the oscillations. In theory, we have $\phi = 0$, but uncertainties in implantation times, broad internal field distributions, and IC orders can cause $\phi \neq 0$, so the parameter was fitted. However, we confirm that the value is close to 0 at the base temperature.

Fitting the temperature dependence of the TF coefficients [Fig. 9] with a sigmoid function, allows us to extract $T_N$ for ZnReO$_4$ as $T_N^{\text{Zn}} = 125.67(10)$ and $T_N^{\text{Mg}} = 83.35(2)$ for MgReO$_4$, where the error represents the variance in temperature from the fitting results.

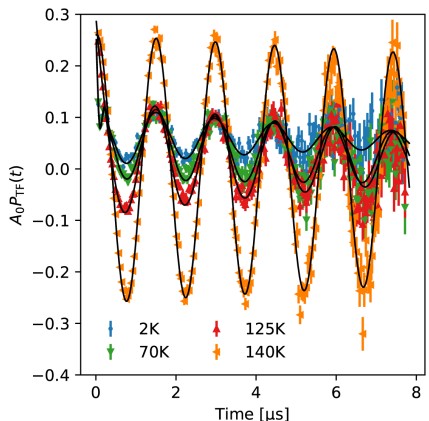

Figure 8: $\mu^+$SR data of MgReO$_4$ in transverse field (TF) of $B = 50\,$G with errorbars and corresponding fits using polarization function (6) in black.

The TF asymmetry is not completely suppressed below $T_N$ and stays $\approx 20\%$ for both compounds until base temperature. This is likely due to impurities in the samples.

Furthermore, we observe a peak in $\lambda_{AF}$ before $T_N$ for ZnReO$_4$. We expect the relaxation rate to increase close to a phase transition with the onset of spin fluctuations associated. Therefore, we expect a cusp at the transition temperature. The reason we do not observe this for MgReO$_4$ could be due to the small AF1 asymmetry fraction.

We also see changes in the asymmetry distribution between the tail and AF component close to $T_N$ for ZnReO$_4$. This indicates that the two components become more difficult to distinguish closer to $T_N$ and could be the cause of the relaxation rate dip before transition.

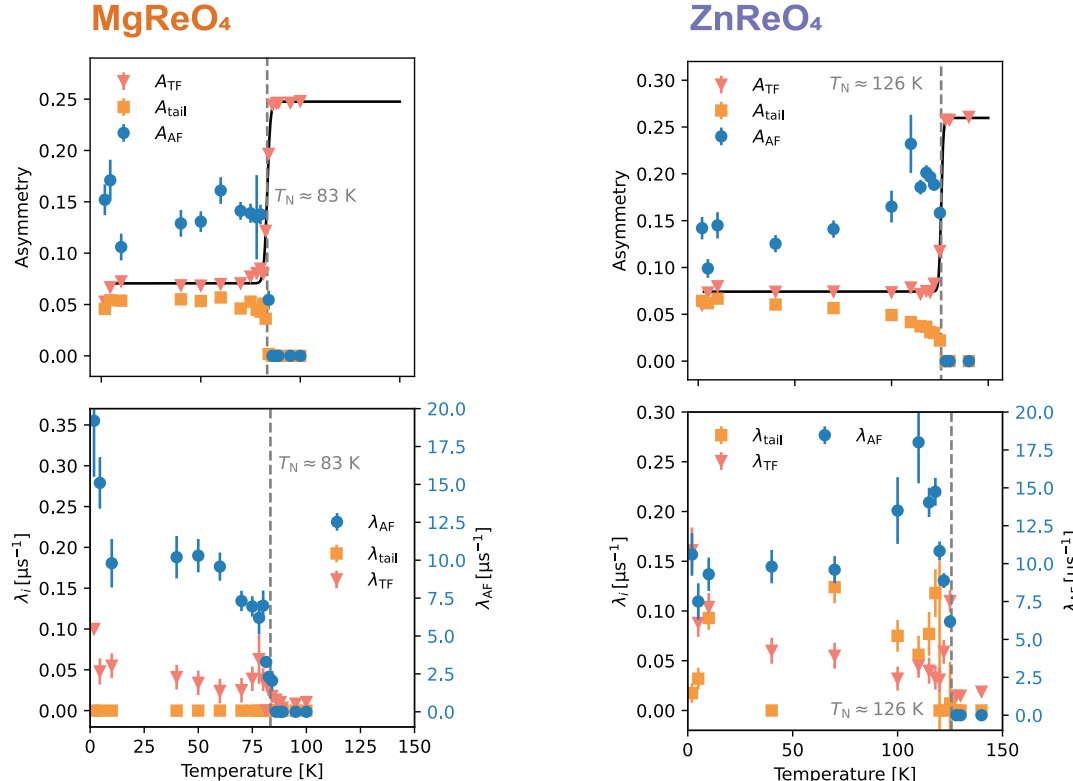

Figure 9: Temperature dependent TF coefficients from polarization function (6) for MgReO$_4$ (left) and ZnReO$_4$ (right). The Néel temperature $T_N^{Zn} = 125.67(10)$ K and $T_N^{Zn} = 83.35(2)$ K are marked in with a grey line. The TF asymmetry is fitted with a sigma function shown in black.

# F   Canting in MgReO4

To examine the possibility of the canting scenario in MgReO$_4$ we let the spins cant within the $ac$ plane using the IRs $\Gamma_3$ and $\Gamma_4$. Fig. 10 shows the evolution of the two muon site frequencies as a function of canting angle. We see that for $\Gamma_3$ the two frequencies never fully meet and are always separated by at least 0.4 MHz. For $\Gamma_4$, the two frequencies meet for the first time at 61° at 6.1 MHz. In our experiments we see that the higher frequency AF2 is no longer distinguishable, but we do not see a drastic increase in the lower frequency, which would be expected if the cause of the reduction in frequencies would be spin canting. Otherwise it would mean a sudden drastic decrease in the magnetic moment exactly coinciding with the lower frequency not changing noticeably. This seems quite unlikely, especially since we do not observe any structural transition in the compound in this temperature range either.

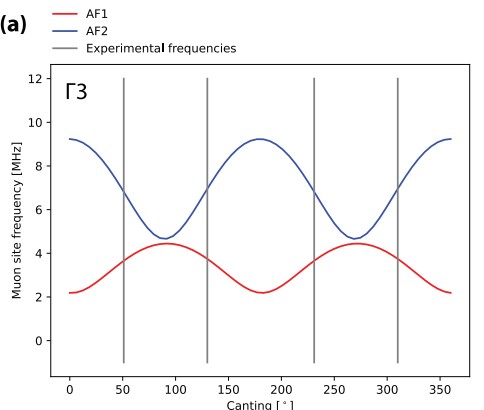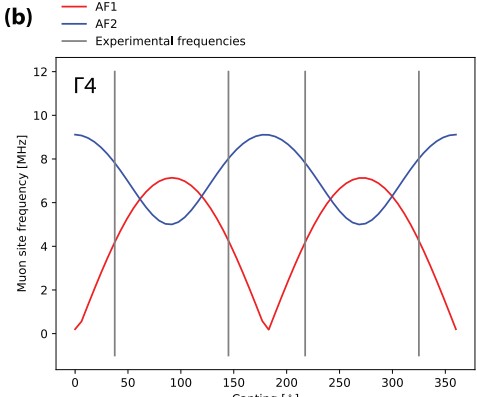

Figure 10: Experienced muon precession frequency and experimentally observed frequency from zero field muon spin spectroscopy $\mu^+$SR data for the two sites of MgReO$_4$ for the magnetic strucures a) $\Gamma_3$ with $\mathbf{k} = (0, 1/2, 0)$ and b) $\Gamma_3$ with $\mathbf{k} = (0, 0, 0)$. Showing the muon sites experiencing the same frequency at a set of angles at around 6 MHz for $\Gamma_4$, but not for $\Gamma_3$

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
