# Peer review of "Characterization of Nuclear and Magnetic Structures of Wolframite-Type MgReO4 and ZnReO4"

_SciPost Physics_

## Round 2 · Referee Report · Anonymous (Referee 1) · 2025-5-15

Report

The present study is concerned with the magnetic ground state properties of AReO4 (A = Mg, Zn), a system in which Re6+ ions decorate a quasi-one-dimensional zig-zag network of distorted ReO6 octahedra along the crystallographic c-axis that are bridged by distorted AO6 octahedra along the other directions. The authors outline a high-pressure synthetic procedure for the preparation of polycrystalline samples, suggest possible magnetic ground states for both compounds and suggest a 6+ valence state of the Re ions. These conclusions are drawn from neutron powder diffraction data, muon+ spin spectroscopy, muon stopping site calculations, and bond valence sum calculations. Overall, I think the musr data were well described and analysed, but beyond that, I do not think that the manuscript merits publication in SciPost Physics, and it would require significant changes for publication in SciPost Physics Core. Some of the reasons follow: 1- The study does not contribute any concrete conclusions or open new pathways for understanding the title compounds. Instead, it mostly presents analysed data with minimal insights. Additionally, as most of the study’s conclusions (musr stopping sites calculations, bond valence calculations) depend on the crystallography done by the authors, more work is necessary to confirm the average structure of the compounds. In a previous study on MgReO4 done by the same group (E. Nocerino et al. J. Phys.: Conf. Ser. 2462 012037), for example, through bulk property measurements, the authors have suggested that the material undergoes a structural transition at 280 K. Could symmetry lowering be the source of the impurity peaks observed in both compounds? And if that is the case, do any of the calculations hold? This could be easily confirmed through a few simulations. 2- The neutron absorption cross section of Re is relatively large, and the sample size (0.5 g) used for the experiment could explain not seeing any magnetic intensity. The subtracted diffraction patterns were also collected at least 80 K apart, which would hide intensities behind artefacts caused by the thermal contraction. The authors could perhaps provide upper estimates on the moment size and peak positions for the suggested magnetic structures. 3- The symmetry analysis for the magnetic structures begins with assuming propagation vectors similar to other 3d metal-based wolframite compounds. This assumption does not hold as the electronic structure of a 5d1 oxide could significantly deviate from that of 3d metals, and the study does not consider any symmetry lowering introducing other possible irreps. 4- It is generally assumed throughout the paper that both systems have a small ordered moment and that this stems from a non magnetic Jeff = 3/2 state with M = L-2S = 0 (L = 1, S = ½). This assumption, however, only holds for a cubic crystal field, and any slight deviations, even in the well ordered Re6+ perovskites (please see Re6+ double perovskites), introduces an appreciable moment (<0.2 mub) that can be seen using PND. This is especially not the case in AReO4 where the octahedra are strongly distorted and manifold mixing is to be expected. The implications of this should be discussed in the manuscript. If the orbital moment is quenched due to this distortion, for example, could the small ordered moment be a result of a 1D Heisenberg chain AFM model? 5- The reasoning for the loss of one of the observed muon precession frequencies in MgReO4 should be expanded on as it is currently unclear. 6- Bond valence sum calculations require very high quality diffraction data and refinement that the study does not have. I’m also surprised that the R0 value obtained for Re6+ is smaller, rather than larger as is usual, than that of Re7+. Some minor points: 1- The current structure figures are unclear, and the zig-zag chains should be clear. 2- New figures are necessary for the diffraction data with zoomed in versions for the difference plots and an explanation for the observed quality of the fits 3- The crystal structure parameter table should be expanded to include the U parameters, occupancies, refinement goodness of fit parameters, etc, with the appropriate error bars that are not currently defined. The bond lengths should also be presented. 4- A list of the temperature dependent parameters for the phase parameter of the TF fits that are not at 0 as one would expect should be presented and an explanation for why these deviate as much as they do should be discussed. 5- The AC susceptibility data in appendix A is inconclusive and is only shown from 40 – 250 K. 6- The impurity phase was presented as a paramagnetic impurity, but is then discussed as a ferromagnet in appendix section A.

Attachment

Recommendation

Reject

---

## Round 2 · Referee Report · Anonymous (Referee 2) · 2025-5-26

Report

The manuscript presents a study of wolframite AReO₄ compounds (A = Mg, Zn) using neutron diffraction and muon spin spectroscopy. These compounds consist of one-dimensional zigzag chains of edge-sharing ReO₆ octahedra, with Re⁶⁺ ions in a 5d¹ electronic configuration. While MgReO₄ was previously reported by the authors, here both compounds are synthesized under high pressure and characterized in polycrystalline form. Neutron and x-ray diffraction indicate the presence of unidentified impurities. The authors use muon spin spectroscopy to elucidate magnetic long-range order yet find no magnetic Bragg peaks in neutron diffraction. They perform muon stopping site calculations to estimate reduced magnetic moments and propose possible magnetic structures, attributing the absence of magnetic reflections in neutron diffraction to the reduced moment. Bond valence sum analysis supports the assignment of the Re oxidation state as 6+.

The study presents interesting new materials and includes a thorough muon spin spectroscopy analysis, as well as muon stopping site and bond valence sum calculations. However, I find that the conclusions regarding the ground states of ZnReO₄ and MgReO₄ are not fully convincing. Additionally, the possible presence of multipolar order—highly relevant in strongly spin-orbit-coupled d¹ or d² systems—is not discussed. Therefore, I do not recommend this manuscript for publication in SciPost Physics. It may become suitable for SciPost Physics Core after revision. My concerns and suggestions are detailed below.

1. Characterisation of ZnReO4
It would be recommended to include DC magnetic susceptibility and specific heat to substantiate the claim of long-range magnetic order in ZnReO4. However, it would appear the air-sensitivity of this material poses significant experimental challenges. The authors report AC-susceptibility shown in Appendix A, Figure 4, but don’t show the temperature range below approx. 35 K. They ascribe the upturn at 100 K is a ferromagnetic impurity similar to what was observed in MgReO4 (E. Nocerino et al., J. Phys.: Conf. Ser. 2462, 012037 (2023), Figure 2 (a)). However, in the case of MgReO4 the upturn at low temperatures is attributed to a paramagnetic tail originating from orphan spins, as discussed in the previous publication. For ZnReO₄, the magnitude of the low-temperature AC susceptibility signal and its interpretation as a ferromagnetic impurity remain unclear and require clarification.

2. Neutron powder diffraction data
The refined neutron powder diffraction patterns in Fig. 1(a) and (b) display several unindexed peaks, and the fits to peak shapes and intensities for the indexed reflections are not ideal. This raises questions about the structural model, sample quality, and potential presence of secondary phases. While it is appreciated the challenges due to high-pressure synthesis and air-sensitivity, these factors may undermine the reliability of the analysis and conclusions. In the high-temperature–low-temperature difference pattern, the inset for ZnReO₄ shows some possible reflections (q=0?), but these are not clearly visible and may be artifacts from thermal shifts.

The authors suggest that the absence of magnetic reflections is due to the reduced moment inferred from muon calculations. However, only 0.5 g of sample was used for neutron powder diffraction. In contrast, G. J. Nilsen et al., Phys. Rev. B 103, 104430 (2021), reported observable magnetic Bragg peaks in Ba₂YReO₆ (Re⁵⁺, 5d²), using 10 g of sample hosting comparable dipole moments (although with polarized neutron diffraction experiment). Therefore, one of the reasons for absence of magnetic reflections could be small amount of sample. The authors should discuss in more detail why magnetic reflections are absent and should provide simulated patterns showing the expected signal for their computed moments and proposed magnetic structures.

3. Muon spin spectroscopy and calculations
The muon spin spectroscopy analysis and simulations are comprehensive, and I have no major concerns about this aspect. Nevertheless, it would strengthen the manuscript to present muon spin spectra of MgReO₄ near the 280 K transition and to discuss expected neutron diffraction peaks based on the proposed magnetic structures. Finally, as mentioned above, it would be recommended to include other experimental results (specific heat, magnetic susceptibility, neutron diffraction with larger sample) to support the obtained magnetic ground states.

4. Context of multipoles
There is a significant amount of literature on multipole physics of 5d transition metal compounds with d1 and d2 configuration (e.g. G. Chen et al., PRB 2010, J. Romhanyi et al., PRL 2017, H. Ishikawa et al., PRB 2019…). In d1 systems, the spin-orbit-entangled Jeff = 3/2 state with g-factor = 0 gives suppressed dipole moment and higher-order multipoles are expected to dominate. Multipolar order is observable with few experimental probes and hence is called “hidden order”.

While the ReO6 in the wolframite compounds in this study are distorted, unquenched orbital moments may persist. For instance, the authors‘ previous work (E. Nocerino et al., J. Phys.: Conf. Ser. 2462, 012037 (2023), Figure 2 (d)) reports a specific heat anomaly at 280 K in MgReO4, which is discussed to be structural in origin; could this correspond instead to multipolar order? The possibility is not addressed in the present manuscript, nor is the broader relevance of multipolar physics to the ground states of MgReO₄ and ZnReO₄.

The authors mention only briefly that the nominal Jeff = 3/2 state could explain the reduced moment of Re6+, but do not fully discuss ground state interpretations or relate their findings to the literature on multipolar physics. I believe this context is important and should be addressed.

5. Other remarks
a. Please provide error bars for the refined values in Tables 1 and 6.
b. Could the authors explain why the χ² values in Table 1 are less than 1?
c. Extracting crystal structure parameters from data on degraded samples (as in Fig. 5) is questionable—the reported oxygen positions appear unreasonable and may be artefactual. I advise caution in interpreting or reporting results from these refinements.
d. Given the moderate refinement quality of the neutron powder diffraction data, the validity of the bond valence sum analysis may be questionable. The authors should comment on the limitations and potential uncertainties introduced by using such data for this analysis.

Attachment

Recommendation

Reject

---

## Editorial Decision

awaiting_resubmission